# Tribochemistry: A Review of Reactive Molecular Dynamics Simulations

**Ashlie Martini [1],\***, **Stefan J. Eder [2,3]** and **Nicole Dörr [2]**

[1] Department of Mechanical Engineering, University of California Merced, 5200 N. Lake Road, Merced, CA 95343, USA

[2] AC2T research GmbH, Viktor-Kaplan-Straße 2/C, 2700 Wiener Neustadt, Austria; Stefan.Eder@ac2t.at (S.J.E.); Nicole.Doerr@ac2t.at (N.D.)

[3] Institute of Engineering Design and Product Development, TU Wien, Getreidemarkt 9, 1060 Vienna, Austria

\* Correspondence: amartini@ucmerced.edu

**Abstract:** Tribochemistry, the study of chemical reactions in tribological interfaces, plays a critical role in determining friction and wear behavior. One method researchers have used to explore tribochemistry is "reactive" molecular dynamics simulation based on empirical models that capture the formation and breaking of chemical bonds. This review summarizes studies that have been performed using reactive molecular dynamics simulations of chemical reactions in sliding contacts. Topics include shear-driven reactions between and within solid surfaces, between solid surfaces and lubricating fluids, and within lubricating fluids. The review concludes with a perspective on the contributions of reactive molecular dynamics simulations to the current understanding of tribochemistry, as well as opportunities for this approach going forward.

**Keywords:** tribochemistry; mechanochemistry; reactive molecular dynamics simulation

---

## 1. Introduction

Chemical reactions at the interface of two sliding surfaces determine tribological behavior in both lubricated and unlubricated systems. The study of these reactions is referred to as tribochemistry. The term tribochemistry has been used relatively flexibly in the literature, but can generally be thought of to encompass chemical reactions that are driven by conditions arising in tribological contacts [1–4]. Tribochemical reactions play a critical role in most tribological applications [5]. As such, tremendous efforts have been and are being made in research and development to optimize friction and reduce wear using approaches based on tribochemical mechanisms. Examples of applications of tribochemistry include anti-wear additives such as zinc dialkyl dithiophosphates [6,7], coating materials such as diamond-like-carbon [8,9], superlubricity phenomena [10,11], new lubricant components such as ionic liquids [12,13] and nanoparticles [14,15], and metalworking processes [16,17]. In most of these technologies, friction and/or wear reduction is attributed to tribofilms that form on surfaces during sliding. It is known that the formation of these films is tribochemical in nature and that shear stress is a necessary condition for film growth [18–20].

Theoretical approaches to studying tribochemical reactions are based on concepts developed within mechanochemistry, the study of chemical reactions that are induced by the direct adsorption of mechanical energy [21,22]. Mechanical energy can be transferred to a chemical system through various types of applied force, e.g., shearing, stretching, and/or grinding [23,24]. In this context, tribochemistry can be thought of as a subset of mechanochemistry [3], where reactions are driven by the shear force that is inherent to a tribological contact. Shear force acting along the reaction coordinate for a given

---

reaction lowers the activation energy and increases the rate of that reaction. The rate of a tribochemical reaction $k$ will increase exponentially with shear force $F$ according to the Bell model [25]:

$$k(F) = A \exp\left(-\frac{\Delta E - F\Delta x^{\ddagger}}{k_B T}\right) , \tag{1}$$

where $\Delta E$ is the thermal activation energy, $\Delta x^{\ddagger}$ is the distance along the reaction coordinate, $k_B$ is the Boltzmann constant, $T$ is temperature, and $A$ is a pre-exponential factor. The first term in the numerator of the exponent corresponds to the energy barrier for a reaction driven thermally and the second term to the amount by which that barrier is lowered by shear force. In tribological studies, this equation is often re-written in terms of the thermal rate constant $k_0 = A \exp\left(-\frac{\Delta E}{k_B T}\right)$ and shear stress $\sigma$ as [4,19,26]:

$$k(\sigma) = k_0 \exp\left(\frac{\sigma \Delta V^{\ddagger}}{k_B T}\right) , \tag{2}$$

where $\Delta V^{\ddagger}$ is called the activation volume [19]. These equations provide a theoretical basis for the tribology community's interpretation of tribochemical processes, but there is still a gap in our understanding of how shear force lowers energy barriers for chemical reactions in sliding contacts, particularly those leading to tribofilm formation.

To understand tribochemical reactions and correlate tribofilm composition with friction and wear behavior, surface characterization has become an indispensable tool. Tribofilm analysis is typically performed ex situ and ex post. For insight at the atomic and molecular level, X-ray photoelectron spectroscopy, X-ray absorption near-edge structure spectroscopy, time-of-flight secondary ion mass spectroscopy, and transmission electron microscopy, among others, are common practice (see, for example, [27–30]). Such ex situ studies have recently been complemented by in situ observations of tribofilm growth at the macroscale by integrating a tribometer in a synchrotron X-ray absorption spectrometer [31,32] and at the nanoscale through use of an atomic force microscope to both mechanically drive film growth and image that growth [33]. Despite the progress made in surface characterization, direct observation of tribochemical processes remains a major experimental challenge since a realistic representation of a tribocontact and the required in situ analysis are mutually exclusive. Partially overcoming this issue, experimental studies have been complemented by atomic-scale simulations that offer a way to "see" inside a tribological interface.

To model tribochemical reactions, atomic-scale simulations must be able to capture the formation and breaking of chemical bonds. Density functional theory (DFT) is a quantum mechanical modeling method that uses functionals of the electron density to explore the electronic structure of systems consisting of atoms and molecules. This approach can be used to calculate energies and chemical structures with relatively high accuracy and has been used to study tribochemical processes (see, for example, [34–36]). However, these approaches are sometimes limited for studying tribochemistry because the computational cost severely limits the size of model systems. In contrast, classical molecular dynamics simulations describe quantum-mechanical effects through approximate, empirical equations. These empirical models are also called force fields or potentials, where the terms are used interchangeably in the literature. Empirical model-based simulations sacrifice some accuracy compared to calculations from quantum mechanical models, but can handle much larger system sizes and are inherently dynamic. However, classical empirical models assume chemical bonds between atoms to be permanent, which means they cannot be used to study chemical reactions. To address this, reactive empirical models have been developed, and the simulations that use these are referred to as "reactive molecular dynamics simulations". Reactive molecular dynamics simulations are often slower than those that use non-reactive potentials because of the smaller simulation time step and more complex functional forms, but their ability to model chemical bonding/de-bonding has enabled their wide application in recent years for studies aimed at understanding the relevant tribochemical processes that determine friction and wear in sliding contacts [37].

At the heart of any molecular dynamics simulation lies the force field that governs how the atoms and molecules in the system interact. For a comprehensive account of the most important reactive force fields, the reader is referred to recent reviews that feature their own application examples from various sub-fields of chemistry or materials science [38–41]. Here, we will give a brief introduction to the force fields that have been applied to study tribochemistry, as described in this review. These force fields are based on the concept of bond order, where the strength of a bond depends on the local chemical environment, as originally developed by Abell [42]. The bond order concept underlies all the reactive potentials that have been used for studies of tribochemistry.

The bond order concept developed by Abell [42], Tersoff [43] and Brenner [44] started as an approximated chemical pseudopotential theory, where the complex core electron movements are smoothed via an effective potential. It was then extended by bond angles and a symmetry concept, finally being able to account for radical formation and the complex dependence of the atomic interactions on local atomic coordination numbers. This led to the so-called first- and second-generation REBO (reactive empirical bond order) potentials [45] that were optimized for carbon-based materials and the AIREBO (adaptive intermolecular REBO) potential [46] that treats the interlayer repulsion in graphite more realistically by introducing non-bonded interactions and torsional parameters. The strength of this family of potentials lies in their accurate chemical description of carbon, including materials relevant to tribology such as hydrocarbons, diamond, and graphite or graphene.

Recent reactive potentials use a variable charge approach to overcome limitations with fixed-charge bond order formalisms [47]. The most prominent examples of this as applied to tribochemistry are the charge-optimized many-body (COMB) potential and reactive force field (ReaxFF). COMB, developed by Sinnott, Phillpot, and co-workers, was originally designed to model Si and silicon dioxide ($SiO_2$) and has been primarily applied to study the tribochemistry of those materials [48,49]. ReaxFF, developed by van Duin and co-workers, uses a bond order formalism with polarizable charge descriptions and includes not only chemical bonds, but also van der Waals and Coulomb interactions, and its long-range bond order terms enable ReaxFF to model transition states where the atoms are farther apart [38,50]. The bond order-based terms and the non-bonded terms are calculated independently so that no a priori distinction between covalent and ionic interactions has to be made. There are now ReaxFF parameters available for many different tribologically relevant materials, and its use has become commonplace in studies of interfacial phenomena and mechanisms, especially modeling reactive chemistry at heterogeneous interfaces.

In this review, we summarize the use of reactive molecular dynamics simulations to study tribochemical processes. The scope of the review is limited to simulations with empirical potentials (i.e., ab initio calculations are not included) that capture the formation and breaking of chemical bonds driven by shear force. The review excludes studies that do not consider sliding, although reactive molecular dynamics simulations have been used to model tribologically relevant reactions driven thermally and/or via normal force (see, for example, [51–56]). Furthermore, although the empirical model used for each study discussed in this review is mentioned, the different potentials are not compared.

Tribochemical systems include solid surfaces and lubricants, so the relevant chemical reactions can occur within and between the two solid surfaces, within the lubricating material, or between the lubricant and solid surfaces. As such, this review is organized in terms of those three interaction categories. First, studies of the formation of bridging bonds between two sliding surfaces or surface reconstruction driven by shear are reviewed. Such studies are relevant to understanding the fundamental mechanisms of friction in general, as well as to specific dry-sliding applications such as solid lubricant coatings and atomic force microscopy. Next, simulations of reactions between lubricants and surfaces are described, for applications including surface polishing, micro-electromechanical systems (MEMS) lubrication, and tribofilm formation reactions. Lastly, studies of reactions that occur within lubricating species, either liquid or gaseous, where shear drives dissociation and/or association

reactions that affect sliding, are reviewed. After individual studies within each topic are presented, the key contributions of reactive molecular dynamics simulations to the scientific community's understanding of fundamental tribochemical processes are summarized. Finally, the review concludes with a discussion of the limitations of reactive molecular dynamics simulation along with suggestions for promising avenues for future research using this method to explore the topic of tribochemistry.

## 2. Review

### 2.1. Reactions between Solid Surfaces

Although many tribological systems rely on liquid lubrication for the most efficient and safest performance, dry friction plays an important role in solid protective coatings, mining, aerospace applications, atomic force microscopy, fail-safe running functions, and others. The most actively studied topic that does not feature an organically-based lubricant is the friction between surfaces and/or tips of nanoscale probes. Many simulations have focused on carbo-based materials such as diamond or diamond-like carbon (DLC), i.e., amorphous carbon, either tetrahedral (ta-C) or hydrogen terminated (a-C:H). Reactive molecular dynamics simulations have also been used to study solid-solid tribosystems consisting of tungsten and tungsten carbide, silicon dioxide, and polytetrafluoroethylene.

In a first set of pioneering studies starting in the 1990s, the Brenner potential [44] was used to study the tribochemistry and wear occurring between diamond surfaces terminated with hydrogen and ethyl groups [57]. It was found that hydrogen atoms sheared from ethyl groups could either recombine with a radical site or abstract another H to form an $H_2$ molecule that remained trapped in the interface. The mechanism of molecular wear debris development was identified as the repeated forming and breaking of bonds between radical sites on the opposing surfaces. A later study sought tribochemical explanations for the friction performance of tribopairs consisting of combinations of H-terminated diamond and H-free amorphous carbon using the REBO force field [58]. It was concluded that films containing a larger fraction of surface $sp^2$-hybridized carbon will exhibit higher levels of friction (compared to $sp^3$) due to the increased adhesion via tribochemical reactions. Up to a typical critical load, a self-mated hydrogenated diamond tribopair exhibited much lower friction than amorphous-C-coated diamond. However, beyond that load, commensurability between the terminating H atoms led to interdigitation that required them to rotate around each other during sliding, thus sharply increasing friction. In the amorphous-C-coated system, a sudden drop in the friction force could be correlated with a decrease in the number of C–C bonds oriented in the sliding direction. A later study confirmed that carbon atoms with unsatisfied valences, irrespective of their hybridization, act as seeds for the formation of interfilm bonds [59]. Snapshots and density profiles from a simulation in which bonds formed across a sliding interface are shown in Figure 1. Such bonding across a sliding interface was shown to increase friction dramatically.

In another set of studies, researchers conducted extensive simulations with hydrogenated DLC coatings using a so-called screened potential based on REBO that was adjusted to describe bond breaking under shear reliably [60]. Shear-driven passivation by methylene groups migrating from one surface to the other, as illustrated in Figure 2, led to significant reduction in friction. Under high loads, hydrogen-rich samples proved to be tribologically superior, featuring less interfacial connectivity that varied little over time, an $sp^2/sp^3$ ratio close to one, and a localized sliding interface, i.e., a well-defined slip plane rather than a region with a Couette-type velocity gradient. One of the mechanisms lowering friction was found to be reorientation of the carbon rings parallel to the sliding interface, rendering dangling bonds less exposed to contact with the counterbody via $sp^2$ hybridization. A subsequent study revealed that polished diamond undergoes an $sp^3$ to $sp^2$ order–disorder transition resulting in an amorphous adsorbed layer that originates from the mechanically-driven dissociation of bonds within the material [61]. This indicates that plastic deformation due to shear occurs mainly in the amorphous phase. Amorphization happens an atom at a time, with so-called "pilot' atoms in an amorphous layer moving over the crystalline surface and exerting time-dependent forces on

the terminating surface atoms. If the pilot atom has a steeper potential energy basin than the surface atom, bonding occurs, which results in removal of the surface atom from its crystalline position. It was concluded that the mechanochemistry of force-driven surface reactions requires a formalism beyond that of Arrhenius activation, which depends on the existence of a stationary equilibrium energy barrier. Using the screened REBO potential [60], it was also found that self-mated diamond and ta-C surfaces formed sp-hybridized carbon chains that led to catastrophic interface failure, especially if the surfaces then came into contact with oxygen [62]. In self-mated a-C:H surfaces, hydrogen atoms in the bulk continuously saturate the chemically active dangling bonds, leading to much smoother interfacial behavior. Recently, it was found that a tribologically-induced hybridization change from $sp^3$ to $sp^2$ can take place on nanocrystalline diamond coatings even under mild tribological conditions [63]. In that study, repeated collisions between diamond asperities led to the formation of a soft $sp^2$+sp amorphous C with interspersed nanodiamond grits ($sp^3$), where grits with {1 0 0} surfaces were particularly prone to decay under shear.

Simulations to study dry friction between tungsten and either tungsten carbide (WC) or DLC surfaces [64–66] used a screened REBO potential for W-C-H systems [67]. In the case of a rough tungsten surface sliding against a similar WC topography, grain refinement was observed on the tungsten side, while the WC counterbody produced a mixed amorphous layer [64]. W/C imbalances at the sliding interface caused discrete atomic events that emitted dislocations into the tungsten bulk, while amorphous shear bands formed within the WC counterbody. For hydrogenated DLC counterbodies, hydrogen diffusion into the tungsten base body was observed due to the distortion of the tungsten lattice, accompanied by transfer of tungsten to the hydrogen-depleted zones of the DLC surface [66]. The high wear could be attributed to the migration of the sliding interface into the tungsten base body where the extent of this migration was determined by the relative order of bond strengths in the system. This model was extended to include hexadecane molecules as a lubricant, as discussed in the next section.

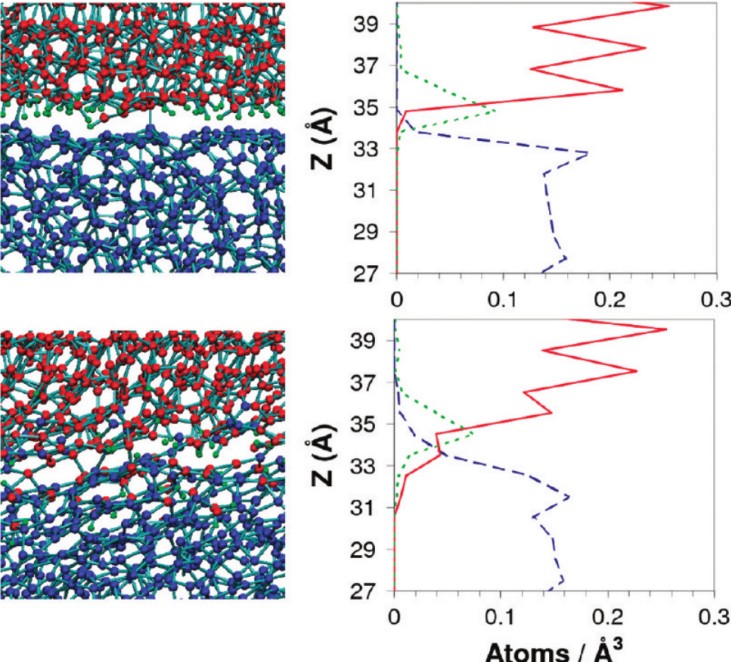

**Figure 1.** Initial (**top**) and final (**bottom**) configurations of a non-hydrogenated carbon film sliding against a hydrogenated counterbody. The associated density plots on the right show the formation of an interfacial transfer film. Colors correspond to: C from the non-hydrogenated surface in blue, C from the hydrogenated counterbody in red, and H in green. From [59]. Reprinted with permission from Schall, J. D.; Gao, G.; Harrison, J. A. Effects of Adhesion and Transfer Film Formation the Tribology of Self-Mated DLC Contacts. J. Phys. Chem. C 2010, 114, 5321–5330. Copyright 2010 American Chemical Society.

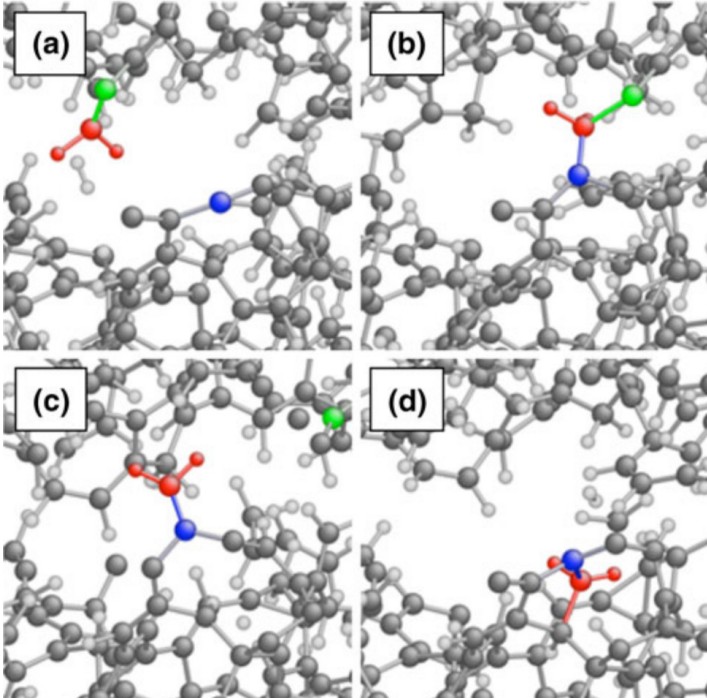

**Figure 2.** Reaction pathway of surface passivation from a simulation of diamond-like carbon (DLC)/DLC sliding. (**a**) the methylene group (red) bound to the upper surface (green) binds to the lower sliding surface (blue) in (**b**). Next, (**c**) shear strains the C–C bonds that connect the methylene to the top and bottom surface until the weaker bond to the top surface finally breaks. Finally, (**d**) the methylene finds a binding position within the bottom surface leading to passivation. Adapted from [60]. Reprinted by permission from Springer Nature Customer Service Centre GmbH: Springer Tribology Letters 39, 49–61, Pastewka, L.; Moser, S.; Moseler, M. Atomistic Insights into the Running-in, Lubrication, and Failure of Hydrogenated Diamond-Like Carbon Coatings. Copyright 2010 Springer.

Reactive molecular dynamics simulations have also been applied to study the wear of silicon and silica. Simulations that employed a modified Tersoff potential with an improved angle-dependent term [68] showed that shear drives amorphization of diamond-cubic Si and diamond [69]. Further, it was suggested that shear-induced amorphous Si is denser than diamond-cubic Si, while amorphous C is less dense than diamond, so the amorphization rates of Si and C exhibit opposite pressure dependence trends. In another study, simulations with the COMB potential were used to complement atomic force microscope experiments of a nanoscale silicon probe with an oxidized surface sliding on an amorphous $SiO_2$ substrate [70]. The simulations underpredicted the volume of probe wear after 40 nm of sliding compared to the experiment, but the trends were consistent. Further, analysis of the crystal structure of the model probe showed that wear was preceded by amorphization, similar to the mechanism proposed previously for diamond [61].

Lastly, a series of studies analyzed the effect of molecular structure, normal load, temperature, and sliding direction on the tribological performance of self-mated polytetrafluoroethylene (PTFE) and polyethylene (PE) surfaces using the REBO potential [71–73]. The two polymeric systems with various crosslink densities had inherently different degrees of structural integrity and rigidity. Wear occurred through multiple processes, including bowing and bunching together of adjacent chains and chain entanglement and scission, as illustrated in Figure 3. In contrast to PTFE, PE chains were not observed to experience chain scission and thus experienced less wear than PTFE. Increasing crosslink density increased the friction force and the coefficient of friction in both materials. Breakage of crosslinks occurred only in the ordered crosslink structures, where chain movement was restricted, leading to greater stiffness. The coefficient of friction remained constantly high (0.5) over the entire temperature range for sliding in a direction perpendicular to the PTFE chain orientation, while it

decreased from 0.2 to 0.1 for parallel orientations. The "violin" configuration, where the chain orientations of the sliding partners were perpendicular to each other, led to a friction coefficient of 0.5 up to temperatures of 100 K and then dropped to a constant 0.3 for 150 K and above. The strongest temperature dependence was observed at zero applied load where adhesion dominated the friction response.

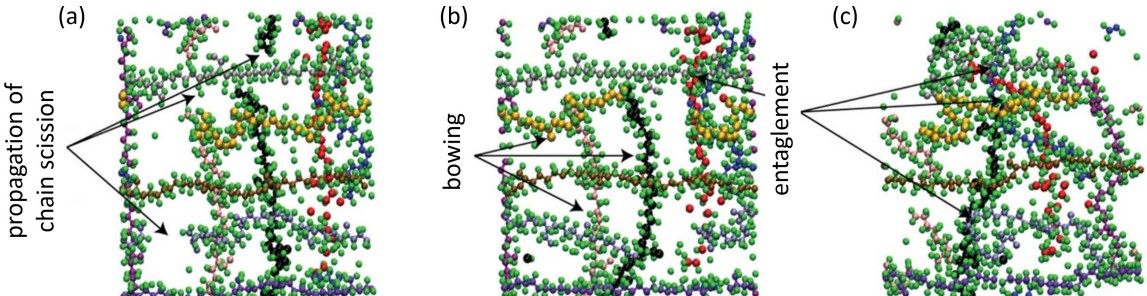

**Figure 3.** Wear simulated in self-mated PTFE contacts occurred through (**a**) chain scission, (**b**) bowing of chains in the shear direction, and (**c**) entanglement of multiple chains. Individual PTFE chains on the surface are differentiated by the color of their carbon atoms; hydrogen is shown in green in all cases. Adapted from [73]. Reprinted by permission from Springer Nature Customer Service Centre GmbH: Springer Tribology Letters 58, 3, 50, Barry, P. R.; Chiu, P. Y.; Perry, S. S.; Sawyer, W. G.; Sinnott, S. B.; Phillpot, S. R. Atomistic Effect of temperature on the friction and wear of PTFE by atomic-level simulation. Copyright 2015 Springer.

In general, reactive molecular dynamics simulations of surfaces sliding relative to one another have revealed several important observations. First, simulations demonstrated the importance of surface hybridization and dangling bonds in determining both friction and wear in dry sliding contacts. Bridging bonds between two surfaces will necessarily increase friction and wear, and simulations with tribologically relevant materials such as DLC suggested avenues for passivating surface atoms during sliding. Furthermore, simulations showed that the orientation of a crystal structure or molecular species relative to the sliding direction is important and can contribute to friction not only through physical resistance to sliding, but also by encouraging or discouraging re-hybridization of surface atoms. Lastly, it has been shown that wear of ordered materials can occur through a two-step process that involves first amorphization and then material removal. These observations are highly relevant to dry sliding applications, especially solid lubricant coatings that may be the only option in scenarios where liquid lubrication is not feasible and tribological behavior is directly determined by the physical and chemical changes on and within the solids during sliding.

## 2.2. Reactions between Lubricants and Surfaces

Chemical reactions between lubricating fluids and solid surfaces are important to many tribological processes. The most common example of this occurs when lubricant additives react with surfaces to form protective films, called tribofilms or tribolayers. Tribofilms are responsible for providing controlled friction and low wear, particularly in boundary lubricated contacts [74]. In other applications, reactions between lubricants and surfaces can be a pivotal part of the surface polishing process. Reactive molecular dynamics simulations have been used to model interactions between lubricants and surfaces to understand how lubricants function in such conditions. However, because many additive molecules are complicated and may contain combinations of elements for which reactive potentials are not available, most previous simulations have focused on model systems.

One series of studies explored shear-driven chemical reactions between water as a lubricant and silicon or silica surfaces [75–78]. This material system is relevant to the super-low friction observed with water-lubricated ceramics, aqueous lubrication of MEMS, as well as chemical-mechanical

polishing processes. ReaxFF simulations of point contact between amorphous $SiO_2$ revealed that water facilitates wear by providing oxygen to form Si–O–Si bonds across the sliding interface [75]. Similar results were obtained from simulations of chemical-mechanical polishing of Si with $H_2O$ or hydrogen peroxide ($H_2O_2$) [79]. These simulations further showed that aqueous $H_2O_2$ led to more oxidation of the Si substrate and in turn faster removal of Si atoms than $H_2O$. Another ReaxFF study of chemical-mechanical polishing on Cu was performed with a silica model pad and a slurry consisting of water, hydrogen peroxide, and glycine [78]. It was found that removal of material from the surface was facilitated by chemisorption of glycine, which reduced the energy barrier for dissociation of Cu atoms from the surface, interactions between $H_2O$ and Cu–OH on the surface, and chemical reactions between the hydroxylated silica surface and Cu substrate, leading to the formation of interfacial Cu–O–Si bridges. However, this increase in wear was not always observed. Another ReaxFF study showed that water can have competing effects on the wear of Si(100) [76]. Specifically, it was shown that water can facilitate wear chemically by providing oxygen to form Si–O–Si bonds and terminal H atoms that help bond rupture, but can also hinder wear mechanically by preventing the contact between the surfaces. Subsequent simulations were performed at a range of temperatures and with varying amounts of water and revealed that both factors affected the formation of covalent bonds that bridge the two solid surfaces [77]. The suppression of wear by water lubricants was also observed in ReaxFF simulations of sodium silicate sliding on amorphous silica [80,81]. Specifically, water was found to inhibit mechanochemical wear by suppressing the formation of $Si_{silica}$–O–$Si_{silicate}$ bonds, consistent with experimental trends showing that wear decreased with increasing humidity [80]. Further, experiments revealed subsurface densification of glass during sliding, which was then explained by the simulations as shear-driven change of the subsurface Si-O-Si bond angle and Si-O bond length distributions [81]. In general, these studies clearly demonstrated the synergy between chemical and mechanical effects in wear of ceramics with aqueous lubrication.

Another set of studies explored reactions between hexadecane and W, WC, or DLC sliding surfaces using a screened REBO potential [60,65,66,82]. Simulations of hexadecane-lubricated W/WC contacts demonstrated that hexadecane forms monolayers that separate the two surfaces and provide low friction because of the low resistance to shear between fluid layers [65]. These observations were consistent with X-ray photoelectron spectroscopy, transmission electron microscopy, and cross-sectional focused ion beam experiments that found that hexadecane lubrication produced less pronounced third-body formation, a thin mixed layer on the WC surface, and only slight grain refinement in the near-surface region of the W material, compared to dry sliding. A similar study with hexadecane lubricating a W/DLC contact pair revealed that the hexadecane also contributed to low friction through chemical means: low shear stress and material transfer were attributed to hexadecane monolayers partially tethered to the a-C:H surface via C–C bonds, which significantly reduced adhesion and resulted in the formation of a layer with low shear resistance consisting of the lubricant and material from both surfaces [66]. More recently, simulations captured hexadecane-lubricated H-passivated diamond sliding against W [82]. This study showed that low friction arose from a low-density hydrocarbon film that formed from dehydrogenation of hexadecane lubricant molecules. The key steps in the reaction leading to this film, illustrated in Figure 4, are hydrogen transfer from the hexadecane to octahedral sites of the tungsten surface, followed by chemisorption of the hexadecyl radical on dangling C-bond sites of the diamond. The formation of carbon-based tribofilms was also captured by ReaxFF simulations of linear olefin chains confined and sheared between metal slabs [74]. The simulations showed that the olefins degrade via dissociation of C–H bonds facilitated by the metal surfaces and scission of backbone C–C bonds. The resulting short-chain dehydrogenated hydrocarbons recombined to form a graphitic tribofilm [74].

Many practical applications of tribology involve ferrous surfaces and recent studies with di-*tert* butyl disulfide, an extreme pressure additive, and Fe(100) surfaces were performed using ReaxFF [51,83]. The reaction pathway was identified as dissociation of the di-*tert*-butyl disulfide (S–S bond breaking), chemisorption of *tert*-butyl sulfide (Fe–S bond formation), and then, *tert*-butyl radical

release (S–C bond breaking), as illustrated in Figure 5a–c, where the radical release was found to be the rate-limiting step [51]. Simulations of sliding across a range of temperatures showed that shear accelerates the reaction and increases the reaction yield [83] The reaction yield was quantified by the rate-limiting step, and the simulations demonstrated that the energy barrier was lowered by shear force acting along the reaction coordinate for the rate limiting step of the reaction. Next, the reaction yield at each temperature was fit to the Bell model [25], Equation (1), rewritten as [83]:

$$\ln(r_y) = \ln(A) - \left(\frac{\Delta E - \Delta E^*}{k_B T}\right) , \qquad (3)$$

where $r_y$ is the reaction yield and $\Delta E^* = F \Delta x^{\ddagger}$ is the amount by which the energy barrier is reduced by shear force. The resultant fit, shown in Figure 5d, enabled $\Delta E^*$ to be calculated from the simulations. These results demonstrated that, at lower temperatures, the yield is larger with shear than without, but the effect of shear was negligible at higher temperatures. This is consistent with findings from previous experimental work that suggested tribofilm formation was thermally driven under extreme pressure conditions (where friction heating is likely to be significant) and driven by shear under moderate conditions (lower frictional heating) [84].

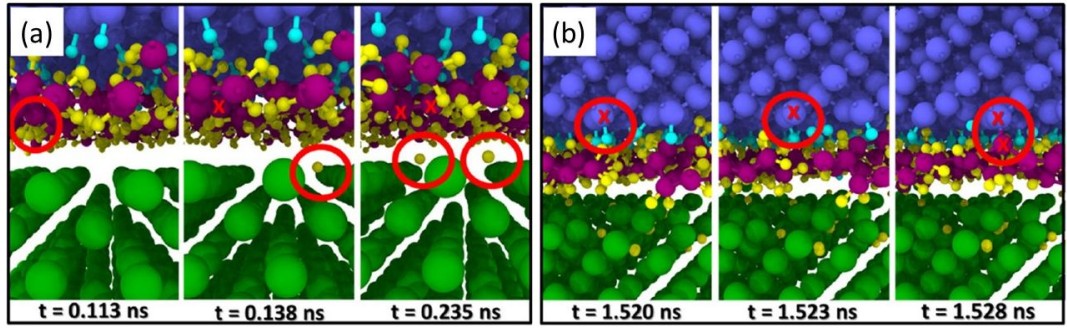

**Figure 4.** Snapshots from reactive molecular dynamics simulations of hexadecane lubricated sliding of tungsten on diamond. Hexadecane lowers friction by physically separating the surfaces, as well as chemically by forming a low-density film on the surfaces. A film forms through (**a**) dissociation of H atoms from hexadecane followed by (**b**) bonding of the hydrocarbon radical with the diamond surface. Atom colors correspond to: diamond C in blue, diamond H in cyan, W in green, hexadecane C in purple, and hexadecane H in yellow. Figure reprinted with permission from the authors of [82].

While the ideal iron surfaces described above may approximate steel on which the native oxide is worn off during sliding, iron oxide models would better represent most tribological interfaces. Unfortunately, there are few reactive force fields available that include the elements necessary to model iron oxide with lubricants or additives. However, recently, parameters for ReaxFF have been developed for interactions between phosphate esters and iron oxide [52,85] and between inorganic alkali polyphosphates and iron oxide [86]. The latter was then used to study sodium pyrophosphate ($Na_4P_2O_7$) confined and sheared between iron(III) oxide ($Fe_2O_3$) surfaces. The simulations revealed shear-driven dissociation of $Na_4P_2O_7$ leading to layering at the liquid–solid interface and diffusion of Na into the surface, in agreement with experimental observations [86].

Reactive molecular dynamics simulations of chemical reactions between lubricants and surfaces have focused on how those reactions affect friction and wear. Studies of water lubrication and silica surfaces are relevant for chemical-mechanical polishing applications, and simulations showed an important synergy between chemical and mechanical effects that led to wear via multiple mechanisms. Hydrocarbon-based liquid lubricants also react with surfaces during sliding. Simulations of hexadecane lubrication showed that even a model base fluid can decrease friction, and the mechanisms were revealed to be chemical reactions that decreased adhesion between the surfaces, as well as the formation of a low shear resistance layer. More recent simulations have focused

on ferrous surfaces that better reflect most tribological surfaces. Simulations of an extreme pressure additive on Fe(100) revealed the critical role that shear force plays in driving the first steps of film formation reactions. Iron oxide was also modeled, and simulations demonstrated reaction pathways for the dissociation of lubricants, as well as how Na diffuses into surfaces to improve tribological performance. Given the importance of tribofilms in general, simulations that reveal pathways for reactions between additives and surfaces have significant potential utility, particularly if they can be used to understand how operating conditions affect film formation rates or suggest avenues for the design of new, optimized additive chemistries.

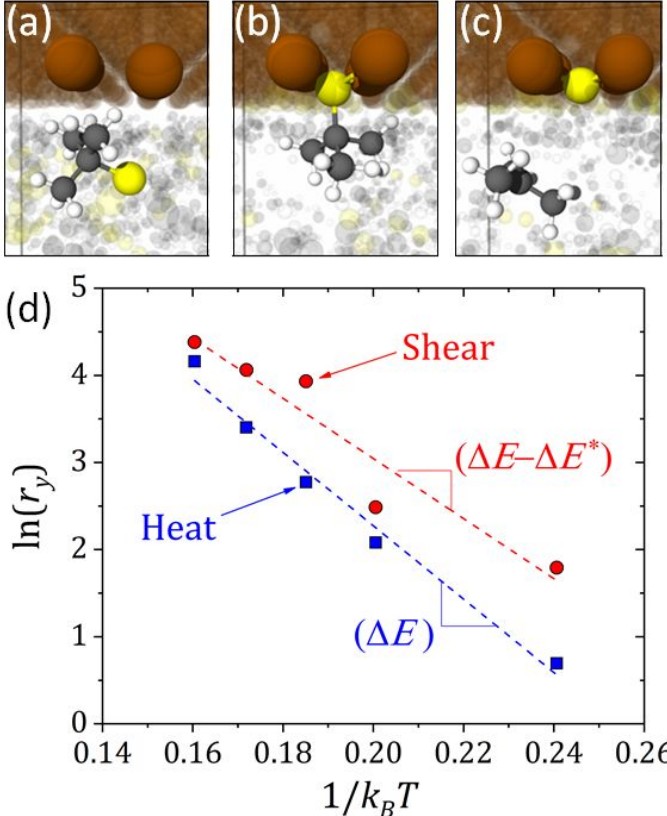

**Figure 5.** Snapshots of a reaction pathway: (**a**) di-*tert*-butyl disulfide approaches the iron surface; (**b**) formation of Fe–S bonds by chemisorption of the *tert*-butyl sulfide radical on the iron surface; and (**c**) breaking of the S–C bond, leading to detachment of a *tert*-butyl radical. Sphere colors correspond to S in yellow, C in gray, H in white, and Fe in brown. (**d**) Natural log of the reaction yield as a function of inverse temperature for the heat and shear stages of the simulation. The data are fitted to Equation (3) such that the fit slope is the reaction energy barrier. When only heat is available to drive the reaction, the energy barrier is $\Delta E$; adding shear decreases that barrier by $\Delta E^*$. Figure adapted from [83]. Adapted with permission from Mohammadtabar, K.; Eder, S. J.; Dörr, N.; Martini, A. Heat-, Load-, and Shear-Driven Reactions of Di-tert-butyl Disulfide on Fe(100). J. Phys. Chem. C 2019, 123, 19688. Copyright 2019 American Chemical Society.

## 2.3. Reactions within Lubricants

In liquid or gas lubricated systems, shear force can drive reactions between the fluid and the confining surfaces (as discussed in the previous section), as well as reactions within the fluid itself. Such reactions can lead to dissociation and/or association of lubricant molecules. Particularly, tribochemical reactions that occur within a fluid are relevant to vapor phase lubrication [87] and to superlubricity observed with some liquid lubricants [11]. In these cases, sliding conditions are typically mild, corresponding to very low flash temperatures such that the chemical reactions are believed to be driven by mechanical force, as opposed to thermal activation. Reactive molecular dynamics simulations have

played a key role in showing how force is transmitted from moving surfaces to a fluid and then how that force drives the breaking and formation of bonds within and between fluid molecules.

Tribochemically-induced oligomerization plays an important role in vapor phase lubrication of MEMS devices [87]. To study this process, ReaxFF simulations were used to study allyl alcohol molecules sheared between amorphous $SiO_2$ surfaces [88]. The resultant oligomerization yield at different normal pressures $P$ was fit to Equation (2), rewritten by replacing shear stress with $\sigma = \sigma_0 + \mu P$ [88]:

$$ln(r_y) \propto \left( \frac{\mu \Delta V^{\ddagger}}{k_B T} P \right) , \qquad (4)$$

where $\sigma_0$ is the shear stress at zero applied force and $\mu$ is the coefficient of friction. It was found that the activation volume obtained from the fitting simulation data was comparable to that measured experimentally for oligomerization of the same gaseous species. The simulations then showed that the reaction pathway induced by shear was different from that observed for thermal reactions [88]. Specifically, shear-driven reactions were found to occur through distortion of the molecules from their equilibrium state, and this distortion occurred when the molecules were covalently anchored to one of the sliding surfaces. The distortion step identified above was also reported in reactive molecular dynamics simulations of alkyne self-assembled monolayers using AIREBO, which showed how crosslinking of chains was initiated by mechanical deformation of the monolayers [89]. Other simulations and experiments of vapor phase lubrication of amorphous $SiO_2$ were carried out with $\alpha$-pinene [90]. This study confirmed the important role of chemisorption by observing more oligomerization on dehydroxylated (more reactive) surfaces. Furthermore, the activation volume extracted from the pressure dependence of the oligomerization yield could be correlated to the distortion step in the reaction pathway. These simulations also demonstrated that shear force, as opposed to normal force, drove these reactions. Together, these findings confirmed a reaction pathway that occurs through $\alpha$-pinene chemisorption, which enables the $SiO_2$ surfaces to transfer shear force to the molecules, followed by distortion of the most highly strained part of the molecule, and finally oligomerization via bond formation between two strained molecules and oxygen from the $SiO_2$ surface; see Figure 6a. ReaxFF simulations of simple geometries showed that the distortion step lowers the energy barrier for the oligomerization reaction, as illustrated in Figure 6b [90].

Superlubricity is the observation of ultra-low friction, typically with coefficients of friction lower than 0.01. One means of achieving superlubricity in a lubricated contact is through the use of a low viscosity liquid that forms layered films with very low shear resistance [11]. One example of a liquid lubricated system for which superlubricity has been observed experimentally is silica lubricated by phosphoric acid [91]. This phenomenon has been studied using ReaxFF of a quartz-quartz contact lubricated by two molecular layers of phosphoric acid that showed friction decreased with increasing temperature, and this trend was correlated with shear-driven oligomerization and the formation of water [92]. The origin of low friction was a slip plane characterized by weak interfacial adhesion where both the oligomers and the water facilitated the formation of this slip plane. Another system with phosphoric acid placed between amorphous $SiO_2$ surfaces also exhibited oligomerization driven by shear [75]. This point contact geometry led to few condensation products in the interface such that oligomerization resulted in higher friction, as shown in Figure 7. The simulations suggested that oligomerization effectively roughened the surface, which facilitated the formation of phosphorus–oxygen bonds across the interface, resulting in higher friction and wear. Superlubricity was also observed in experiments and simulations of ta-C lubricated by glycerol [93]. The simulations using ReaxFF showed that shear drove the decomposition of glycerol to release water, which then formed hydrogen bonds with the surface; the tribodegradation of glycerol was corroborated by experimental evidence. The simulations then suggested that low friction was enabled by slip between the resultant OH-terminated surfaces. However, from an application perspective, tribodegradation of glycerol is likely to be detrimental for steel-on-steel sliding because of corrosion, but could be beneficial for ta-C-coated surfaces.

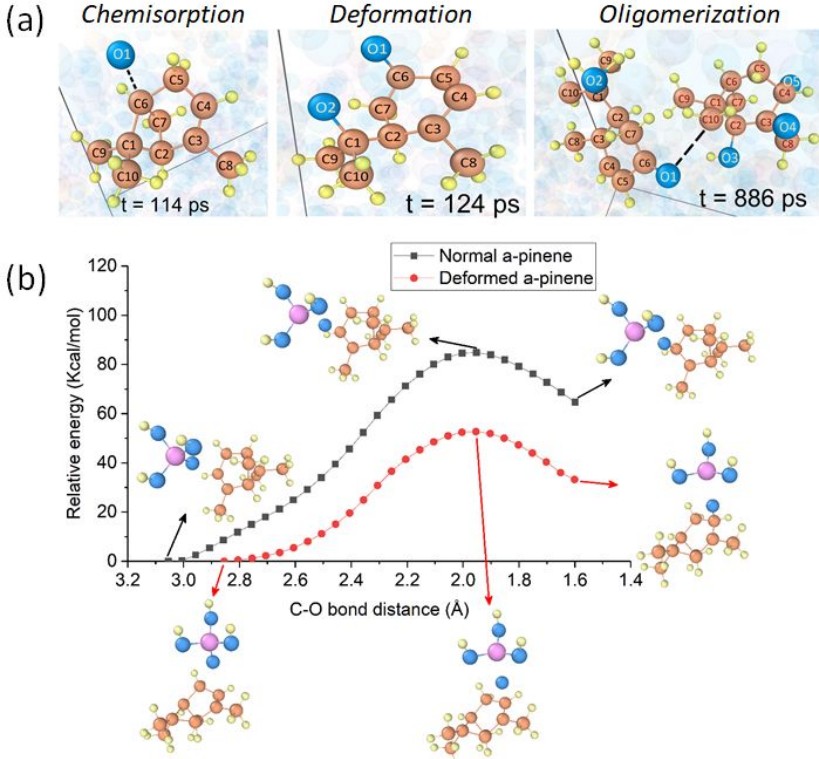

**Figure 6.** (**a**) Snapshots from simulations of oligomerization of α-pinene, which illustrate the key steps of the process: chemisorption, deformation, and finally oligomerization. (**b**) The energy barrier is lower for the intermediate deformed by shear (red) compared to the fully-relaxed, undeformed structure (black). Sphere colors correspond to C in brown, O in blue, Si in pink, and H in yellow. From [90]. Adapted with permission from Khajeh, A.; He, X.; Yeon, J.; Kim, S. H.; Martini A. Langmuir 2018, 34, 5971–5977. Copyright 2018 American Chemical Society.

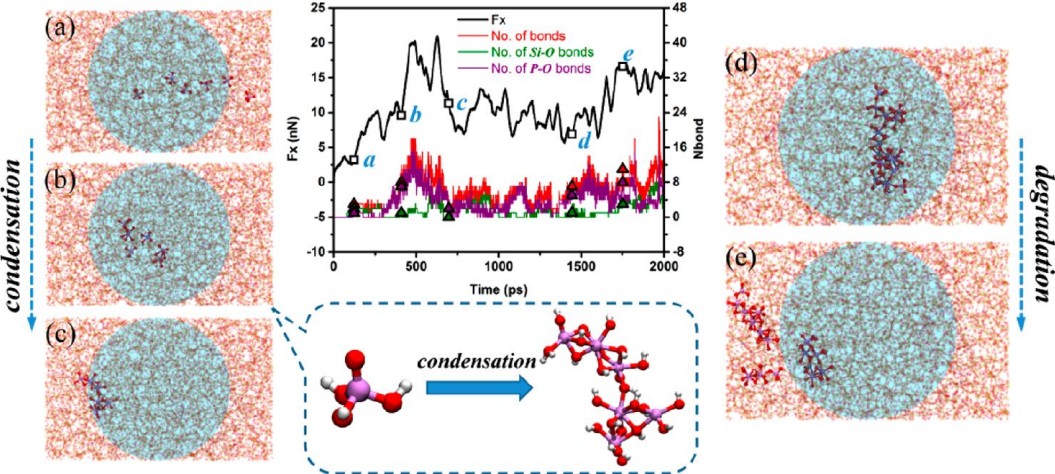

**Figure 7.** Top views of a simulation of an amorphous $SiO_2$ point contact lubricated with phosphoric acid showing oligomerization of phosphoric acid (**a**), (**b**), (**c**) and degradation of its condensation products (**d**), (**e**). Only molecules that take part in the tribochemical reactions are represented as solid; all the others are made semitransparent. The semitransparent blue circles indicate the location of the contact zone. The friction and number of bonds formed between the two sliding surfaces corresponding to the simulation snapshots are also reported (middle plot). From [75]. Reprinted with permission from Yue, D.-C.; Ma, T.-B.; Hu, Y.-Z.; Yeon, J.; van Duin, A. C. T.; Wang, H.; Luo, J. Tribochemical Mechanism of Amorphous Silica Asperities in Aqueous Environment: A Reactive Molecular Dynamics Study. Langmuir 2015, 31, 1429–1436. Copyright 2015 American Chemical Society.

Simulations of chemical reactions that occur within lubricants have shown that such reactions can influence tribological behavior. Importantly, these simulations provide a means of exploring reactions where shear initiates reactions by opening reaction pathways not easily accessible thermally, as demonstrated with oligomerization reactions. Simulations revealed a pathway in which shear deforms a molecule to make it more reactive and thus facilitates oligomerization reactions. From an application perspective, oligomerization has been identified as a main mechanism in water lubrication of MEMS. More importantly perhaps, these studies provide important insights into how shear drives reactions in general and the role that water plays in particular. The latter is attributed to the fact that water is omnipresent, and therefore, it can be assumed that water from the environment is involved in tribochemical reactions. As demonstrated by simulations, water can dissociate during sliding, leading to reaction products that form a low-friction layer. These studies reveal pathways for achieving superlubricity that may be leveraged as scientists and engineers continually seek to lower friction to achieve more energy efficient lubricated components.

## 3. Challenges and Opportunities

As demonstrated by this review, reactive molecular dynamics simulations provide researchers with a tool to explore the reaction pathways and fundamental mechanisms of tribochemical processes. However, as with all scientific techniques, there are limitations to this approach. The key limitations of reactive molecular dynamics simulations for studying tribochemistry are their short length and time scales, inconsistent availability of reactive potential parameters for systems of interest to tribologists, and a lack of direct experimental validation of simulation results.

First, molecular dynamics simulations are typically limited to tens or hundreds of nanometers because of their explicit description of atoms. The size of a simulation can be increased somewhat by running massively parallel simulations on large research computing resources, but an atomistic simulation approach cannot fully account for factors such as surface roughness or elastic deformation that are micro- to milli-scale in mechanical components. Timescale is also a challenge since molecular dynamics simulations are typically limited to tens of nanoseconds by an integration time step that is on the order of femtoseconds. This issue is exacerbated for reactive molecular dynamics simulations that require even smaller time steps (0.1 to 0.25 fs) to resolve the vibrational modes of bonds between atoms. These short durations introduce challenges when trying to model tribochemical reactions that have relatively high activation energies, making them inaccessible to the timescale of atomistic simulations run at moderate temperatures. The brute force way to address this challenge is to make reactions happen faster by running simulations at a higher temperature or faster sliding speed than a corresponding experimental measurement. This approach is widely used and is reasonable, assuming that the reaction pathway does not change as the temperature or driving force is increased. Another approach is to use accelerated molecular dynamics methods that use numerical tricks to encourage reactions to happen sooner. The timescale of reactive molecular dynamics simulations has been extended with multiple accelerated simulation methods [94–97]. Most of these methods are limited to physical phenomena that occur through rare events [98], such as stick-slip friction where very fast slip events are separated by relatively long periods of stick [99] but they do not offer any speed-up for continuous processes. Further, they are based on assumptions that must be thoroughly validated before they can be applied directly to speed up a given simulation. Regardless, the use of accelerated methods with reactive molecular dynamics simulations is a promising approach to modeling slower tribochemical processes under physically realistic conditions. Reactive molecular dynamics simulations that can achieve longer times and therefore be used to model high activation energy barrier reactions will be powerful tools to improving the community's understanding of tribochemical processes. More generally, spatial and temporal multi-scaling techniques are expected to complement computational resources that continue to increase in size and speed to enable atomistic simulations to capture more and more application relevant tribological phenomena.

Second, before a reactive molecular dynamics simulation can be used to model a given material or chemical system, parameters for the empirical potential must be available for the relevant elements and reactions. Most of the reactive potentials have robust parameter sets available for chemistries comprised of carbon, hydrogen, and oxygen. For other elements, however, parameters may or may not be available, depending on the potential formalism and the combination of elements of interest. Unfortunately, most tribologically relevant systems contain many different elements in combinations that are not available in existing parameter sets. For example, reactions between zinc dialkyl dithiophosphate (ZDDP) and ferrous surfaces are critical to tribofilm formation and so would be of interest to model, but at this point, there is no reactive potential parameter set that contains the necessary combination of Zn, S, P, C, H, O, and Fe. The development of new reactive potential parameters is both challenging and time consuming, so most users of reactive molecular dynamics simulations are not able to develop new parameters as they are needed for a given study. However, new parameter sets continue to be developed, so the adverse effect of parameter set availability is decreasing over time. Further, researchers have begun applying advanced techniques such as genetic algorithms to facilitate the parameterization process [100–103]. As additional parameter sets become available, reactive molecular dynamics simulations of more and more complex chemical systems will be possible. For example, recently developed parameter sets enable the simulation of tricresyl phosphate [52] and tetrasodium pyrophosphate [86] on iron oxide. Similar development of potentials for ZDDP or other additives, particularly with ferrous materials, will be extremely valuable. Going forward, the research community should focus on improving the accessibility and usability of parameter set development tools for non-experts, make all new parameter sets publicly available (e.g., as supporting information with the paper reporting the parameterization), and report the range of applicability and limitations of newly developed parameters. The availability of an increasing number of widely available and well-documented reactive potentials will enable reactive molecular dynamics to be applied by more researchers for a wider range of tribologically relevant systems.

Lastly, there remains a gap between results obtained from reactive molecular dynamics simulations and those from experimental measurements, which often precludes direct validation of the models. The time/length scale issue mentioned above is partially responsible for the simulation-experiment gap. However, even for simulations and experiments performed at different scales, partial validation can be obtained by comparison of trends or extrapolation. For example, the pressure dependence of shear-driven polymerization was shown to be similar for reactive molecular dynamics simulations and ball-on-disk experiments, despite the orders of magnitude difference in length scale [90]. In another example, reactive molecular dynamics simulations showed that hexadecane formed monolayers that separated surfaces, consistent with experiments of hexadecane lubrication that exhibited less wear than dry sliding [67]. Qualitative agreement with experiment, although not definitive evidence of model accuracy, provides a level of confidence that the simulation results are realistic from a tribological point of view. Then, once corroborated, the simulations can be used to provide information about a tribochemical process that is not accessible to an experiment, e.g., reaction intermediates. However, such comparisons are difficult to perform if the simulations and experiments are designed and performed separately. Instead, simulations and experiments can be planned together such that they capture the same tribological system, within the limitations of both methods. Finally, it is critical that the repeatability of simulation results be evaluated with the same level of scrutiny as a corresponding experiment. For example, multiple independent simulations should be run so that results can be reported with error bars or confidence intervals. Simulations should be designed to match an experiment and then used to generate results with known accuracy and repeatability that are directly compared to experimental results. Once validated, such simulations can be used to make predictions about properties, e.g., reaction rates for tribofilm formation, across a wide range of operating conditions that then guide a smaller number of experimental studies performed at conditions suggested by the simulations.

## 4. Conclusions

Overall, this review summarized reactive molecular dynamics simulations that have been performed to study shear-driven chemical reactions relevant to the field of tribology. The utility of this simulation tool was demonstrated for dry, as well as lubricated sliding contacts. These contacts necessarily include surfaces and lubricants, and chemical reactions that occur within and between both of these are critical to the friction and wear behavior. In some cases, comparisons between simulations and experimental results were made, and consistent trends were observed. Such comparisons are incredibly important since they provide partial validation of the simulations and demonstrate the relevance of the approach to the field of tribology. Some simulations used model systems and therefore did not have a direct experimental counterpart. However, these simulations are still useful because they explain mechanisms that cannot be measured or observed any other way. Notably, many studies discussed in this review reported reaction pathways that revealed how shear force drives chemical reactions. This understanding will be critically important as tribology moves from a "trial-and-error" approach to knowledge-based design of materials and lubricants. As empirical potentials become more accurate and more widely available and computational resources become faster, reactive molecular dynamics simulations will play a key role in this new concept of tribological design. We anticipate that reactive molecular dynamics simulations will become an essential part of developing both a fundamental understanding of tribological processes and optimization of materials, lubricants, and components, as well as the ability to predict the useful lifetime of tribologically stressed components.

**Author Contributions:** Conceptualization, A.M.; Writing, review and editing, A.M., S.J.E., N.D. All authors have read and agreed to the published version of the manuscript.

**Funding:** This work was supported by the Austrian COMET-Program (K2 projects X Tribology, No. 849109, and InTribology, No. 872176) and carried out at the "Excellence Centre of Tribology" (AC2T research GmbH). The government of Lower Austria supported the endowed professorship in tribology at the TU Vienna (Grant No. WST3-F-5031370/001-2017).

**Acknowledgments:** Final manuscript review and proofreading provided by M. R. Vazirisereshk.

**Conflicts of Interest:** The authors declare no conflict of interest.

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
