# Peer review of "Tribochemistry: A Review of Reactive Molecular Dynamics Simulations"

_lubricants, doi:10.3390/lubricants8040044_

Round 1

Reviewer 1 Report

This is a good review, which provides a perspective view of tribochemistry studies used reactive molecular dynamics (MD) simulations for chemical reactions between counter surfaces, surfaces and lubricant molecules, as well as within lubricants. The review begins with a detailed and comprehensive introduction about different kinds of reactive potentials available at moment, for instance Brener, REBO, AIREBO, COMB, and ReaxFF, as well as the developing history of these force fields. The content is well written with a consistent theme and logic connection between each section. Each statement also has sufficient supporting references. Additionally, this work also highlights some main trends in tribological field, such as oil-based or hydrocarbon lubricant, water/aqueous lubricant, carbon based materials (DLC, amorphous carbon, hydrated/dehydrated DLC…). Several drawbacks in reactive MD simulations are also mentioned in the end of this review.

I recommend a minor revision. However, the following comments should be addressed.

This review mainly focuses on systems in which the lubricant/surface contain C/H/O elements such as diamond, DLC, hydrocarbon, PTFE, PE, glycerol, and water confined between some common Si/SiO2/Cu/Fe/Fe2O3 surfaces. This could be due to the availability of ReaxFF parameters for these other systems, which has been mentioned in the Challenges and Opportunities. Although this review has mentioned some other lubricants such as glycine, di-tert butyl disulphide, and phosphoric acid, but the content is still limited. “If possible”, this review should mention more about the carbon-based chemical tribofilm formation, and 2D material such as Graphene, Graphene oxides, h-BN, black phosphorous (BP) and MoO2 which is a hot topic in tribology at moment, and the ReaxFF parameters for these additive/lubricant have been developed. For more information about these materials, the authors can search them from Adri van Duin’s publication.

Currently, there are also a few reactive MD simulations about carbon-based tribofilm using ReaxFF such as:

  • Erdemir, A., G. Ramirez, O.L. Eryilmaz, et al., Nature, 2016. 536(7614): p. 67-71.
  • Xu, J., J. Nian, P. Wang, Z. Guo, and W. Liu, The Journal of Physical Chemistry C, 2019. 123(3): p. 1677-1691.

And at elevated lubrication:

  • Recently, a new ReaxFF has also been developed for alkali polyphosphate additive confined between iron oxide surface at elevated temperature (Ta, D.T., H.M. Le, A.K. Tieu, S. Wan, A. v. Duin. et al., ACS Applied Nano Material, 2020, Reactive Molecular Dynamics Study of Hierarchical Tribochemical Lubricant Films at Elevated Temperatures)

Author Response

This is a good review, which provides a perspective view of tribochemistry studies used reactive molecular dynamics (MD) simulations for chemical reactions between counter surfaces, surfaces and lubricant molecules, as well as within lubricants. The review begins with a detailed and comprehensive introduction about different kinds of reactive potentials available at moment, for instance Brener, REBO, AIREBO, COMB, and ReaxFF, as well as the developing history of these force fields. The content is well written with a consistent theme and logic connection between each section. Each statement also has sufficient supporting references. Additionally, this work also highlights some main trends in tribological field, such as oil-based or hydrocarbon lubricant, water/aqueous lubricant, carbon based materials (DLC, amorphous carbon, hydrated/dehydrated DLC…). Several drawbacks in reactive MD simulations are also mentioned in the end of this review. I recommend a minor revision.

We appreciate the positive feedback and recommendation for publication after minor revision.

However, the following comments should be addressed. This review mainly focuses on systems in which the lubricant/surface contain C/H/O elements such as diamond, DLC, hydrocarbon, PTFE, PE, glycerol, and water confined between some common Si/SiO2/Cu/Fe/Fe2O3 surfaces. This could be due to the availability of ReaxFF parameters for these other systems, which has been mentioned in the Challenges and Opportunities. Although this review has mentioned some other lubricants such as glycine, di-tert butyl disulphide, and phosphoric acid, but the content is still limited. “If possible”, this review should mention more about the carbon-based chemical tribofilm formation, and 2D material such as Graphene, Graphene oxides, h-BN, black phosphorous (BP) and MoO2 which is a hot topic in tribology at moment, and the ReaxFF parameters for these additive/lubricant have been developed. For more information about these materials, the authors can search them from Adri van Duin’s publication.

We did a thorough literature review and found several more recent papers that use ReaxFF to study shear-driven reactions. They have been incorporated into the revised manuscript.

Currently, there are also a few reactive MD simulations about carbon-based tribofilm using ReaxFF such as:

Erdemir, A., G. Ramirez, O.L. Eryilmaz, et al., Nature, 2016. 536(7614): p. 67-71.

Thank you for bringing this recent reference to our attention. We have incorporated discussion of this paper into the review in the section entitled Reactions between Lubricants and Surfaces.

Xu, J., J. Nian, P. Wang, Z. Guo, and W. Liu, The Journal of Physical Chemistry C, 2019. 123(3): p. 1677-1691.

This paper by Xu et al. reports molecular dynamics simulations performed primarily with the non-reactive COMPASS force field, which are out of scope of our review paper. There were a few ReaxFF-based simulations performed in the study, but the results contradicted experimental observations and it was concluded that there was an issue with the force field, so all subsequent simulations were carried out using COMPASS. Therefore, we did not deem this study to be within the scope of our review paper and have not added it to the revised manuscript.

And at elevated lubrication:

Recently, a new ReaxFF has also been developed for alkali polyphosphate additive confined between iron oxide surface at elevated temperature (Ta, D.T., H.M. Le, A.K. Tieu, S. Wan, A. v. Duin. et al., ACS Applied Nano Material, 2020, Reactive Molecular Dynamics Study of Hierarchical Tribochemical Lubricant Films at Elevated Temperatures)

Thank you for bringing this recent reference to our attention. We have incorporated discussion of this paper into the review in the section entitled Reactions between Lubricants and Surfaces.

Reviewer 2 Report

This is the comments on the paper N ID: lubricants-lubricants-743966

Type of manuscript: Review.

Title: Tribochemistry: A Review of Reactive Molecular Dynamics Simulations.

Authors: Ashlie Martini, Stefan Eder, Nicole Dörr.

Selected Papers from the 7th European Conference on Tribology (ECOTRIB2019) submitted to “Lubricants”.

Title:  Triboemission of fine and ultrafine aerosol particles: a new approach.

for measurement and accurate quantification.

Journal: Lubricants.

This manuscript has been quickly reviewed by reviewer and the comments are attached at the bottom.

Rate the Manuscript:

Significance to field and specialization of “Lubricants” journal:  perfect.

  1. The review summarized reactive molecular dynamics simulations that have been performed to study shear-driven chemical reactions in tribology.
  2. The utility of this simulation tool was demonstrated for dry as well as lubricated sliding contacts. These contacts necessarily include surfaces and lubricants, and chemical reactions that occur within and between both of these are critical to the friction and wear behaviour.
  3. Comparisons between simulations and experimental results were made and consistent trends were observed. Such comparisons are incredibly important since they provide partial validation of the simulations and demonstrate relevance of the tribology approach.
  4. Some simulations studied model systems and therefore did not have a direct experimental counterpart. However, these simulations are still useful because they explain mechanisms that cannot be measured or observed any other way.
  5. Notably, many studies discussed in this review reported reaction pathways that revealed how shear force drives chemical reactions. This understanding will be critically important as tribology moves from the `trial-and-error' approach to knowledge-based design of materials and lubricants.
  6. As empirical potentials become more accurate and more widely available, and computational resources become faster, reactive molecular dynamics simulations will play a key role in this new concept of tribological design. We anticipate that reactive molecular dynamics simulations will become an essential part of developing both a fundamental understanding of tribological processes and optimization of materials, lubricants, and components as well as the ability to predict the useful lifetime of tribologically stressed components.

Originality: good.

  1. Clarity and presentation:  acceptable.
  2. Appropriateness for Journal: appropriate subject mater for the journal “Lubricants”.
  3. Need for rapid publication: yes.
  4. Recommendations: to sent after minor revision to “Lubricants”.

However, the following remarks and questions were arisen after reading of the manuscript:

  1. Abstract should include only main statements and conclusions of study. Please, re-write this section in more compact and clear view.
  2. Section “CONCLUSIONS”: Text should be improved.
  3. All abbreviations should be explained.
  4. Please provide the appropriate references for some equations, because the readers may indentify them as ‘written in first time’.
  5. For the Review article the literature review is may be weak with limited experimental results for comparison of theoretical model. May be consider the next new papers, which consist surfaces and liquid, solid state (powder) lubricants, and chemical reactions that occur during hydrogenation and  critical to the friction and wear in contact zone:
  6.  Specific Features of the Fracture of Hydrogenated High-Nitrogen Manganese Steels Under Conditions of Rolling Friction.- Materials ScienceVolume 50, Issue 4, 1 January 2015, Pages 604-611. DOI: 10.1007/s11003-015-9760-9;
  7. Tribotechnical properties of nitrogen manganese steels under rolling friction at addition of (GaSe)xIn1-x, powders into contact zone.- Metallofizika i Noveishie Tekhnologii, Volume 32, Issue 5, May 2010, Pages 685-695; ISSN: 10241809;
  8. Triboengineering properties of austenitic manganese steels and cast irons under the conditions of sliding friction. Materials Science, Volume 41, Issue 5, September 2005, Pages 624-630. DOI: 10.1007/s11003-006-0023-7.

Final conclusion. All components of the manuscript should be carefully checked: text, figures, equations and tables. Revised version of paper should be prepared and submitted again.   16.3.2020.

Author Response

The review summarized reactive molecular dynamics simulations that have been performed to study shear-driven chemical reactions in tribology.

The utility of this simulation tool was demonstrated for dry as well as lubricated sliding contacts. These contacts necessarily include surfaces and lubricants, and chemical reactions that occur within and between both of these are critical to the friction and wear behaviour.

Comparisons between simulations and experimental results were made and consistent trends were observed. Such comparisons are incredibly important since they provide partial validation of the simulations and demonstrate relevance of the tribology approach.

Some simulations studied model systems and therefore did not have a direct experimental counterpart. However, these simulations are still useful because they explain mechanisms that cannot be measured or observed any other way.

Notably, many studies discussed in this review reported reaction pathways that revealed how shear force drives chemical reactions. This understanding will be critically important as tribology moves from the `trial-and-error' approach to knowledge-based design of materials and lubricants.

As empirical potentials become more accurate and more widely available, and computational resources become faster, reactive molecular dynamics simulations will play a key role in this new concept of tribological design. We anticipate that reactive molecular dynamics simulations will become an essential part of developing both a fundamental understanding of tribological processes and optimization of materials, lubricants, and components as well as the ability to predict the useful lifetime of tribologically stressed components.

Recommendations: to sent after minor revision to “Lubricants”.

We appreciate the positive feedback and recommendation for publication after minor revision.

Abstract should include only main statements and conclusions of study. Please, re-write this section in more compact and clear view.

We have attempted to make the abstract more compact and to the point by removing two of the introductory sentences.

Section “CONCLUSIONS”: Text should be improved.

We reviewed the Conclusions section and did not find any issues with the text. If the reviewer has specific suggestions, however, we would be happy to incorporate them into the manuscript.

All abbreviations should be explained.

We checked all abbreviations and found that all were defined in the paper. They are the following:

  • density functional theory (DFT)
  • reactive empirical bond order (REBO)
  • adaptive intermolecular REBO (AIREBO)
  • charge-optimized many-body (COMB)
  • reactive force field (ReaxFF)
  • micro-electromechanical systems (MEMS)
  • diamond-like carbon (DLC),
  • tetrahedral amorphous carbon (ta-C)
  • hydrogen terminated amorphous carbon (a-C:H)
  • polytetrafluoroethylene (PTFE)
  • polyethylene (PE)
  • zinc dialkyl dithiophosphate (ZDDP)

However, perhaps the reviewer was referring to chemical names. Therefore, we have defined the following acronyms in the revised manuscript:

  • tungsten carbide (WC)
  • hydrogen peroxide (H2O2)
  • sodium pyrophosphate (Na4P2O7)
  • iron (III) oxide (Fe2O3)

Please provide the appropriate references for some equations, because the readers may indentify them as ‘written in first time’.

We have explicitly placed the appropriate reference(s) before each equation.

For the Review article the literature review is may be weak with limited experimental results for comparison of theoretical model. May be consider the next new papers, which consist surfaces and liquid, solid state (powder) lubricants, and chemical reactions that occur during hydrogenation and critical to the friction and wear in contact zone:

 Specific Features of the Fracture of Hydrogenated High-Nitrogen Manganese Steels Under Conditions of Rolling Friction.- Materials ScienceVolume 50, Issue 4, 1 January 2015, Pages 604-611. DOI: 10.1007/s11003-015-9760-9;

Tribotechnical properties of nitrogen manganese steels under rolling friction at addition of (GaSe)xIn1-x, powders into contact zone.- Metallofizika i Noveishie Tekhnologii, Volume 32, Issue 5, May 2010, Pages 685-695; ISSN: 10241809;

Triboengineering properties of austenitic manganese steels and cast irons under the conditions of sliding friction. Materials Science, Volume 41, Issue 5, September 2005, Pages 624-630. DOI: 10.1007/s11003-006-0023-7.

We appreciate the reviewer’s suggestion of incorporating more experimental results into the review. We did attempt to mention correlations between experiments and simulations, were appropriate. However, a comprehensive review of the experimental tribochemistry literature is far out of scope of this paper. With respect to the specific references mentioned above, we did not find direct correlations between these experimental results and any published reactive molecular dynamics simulations. Therefore, we have not incorporated the suggested references into the revised manuscript.

Final conclusion. All components of the manuscript should be carefully checked: text, figures, equations and tables. Revised version of paper should be prepared and submitted again.  

We checked all text, figures and equations again and made minor edits to a correct any outstanding issues.